# Patterns of regulatory behavior in the still-face paradigm at 3 months: A comparison of Brazilian and Portuguese infants

**Marina Fuertes**[1,2]*, **Camila da Costa Ribeiro**[3], **Miguel Barbosa**[4], **Joana Gonçalves**[5], **Ana Teresa Teodoro**[3], **Rita Almeida**[6], **Marjorie Beeghly**[7], **Pedro Lopes dos Santos**[1], **Dionísia Aparecida Cusin Lamônica**[3]

**1** Centro de Psicologia of University of Porto, Oporto, Portugal, **2** ESELX_IPL, Lisboa, Portugal, **3** Departamento de Fonoaudiologia da Faculdade de Odontologia de Bauru, University of São Paulo, São Paulo, Brazil, **4** IFaculdade de Medicina, Universidade de Lisboa, Lisbon, Portugal, **5** Faculdade de Psicologia e Ciências da Educação of University of Porto, Oporto, Portugal, **6** Faculdade de Psicologia of University of Lisbon, Lisbon, Portugal, **7** Department of Psychology, Wayne State University, Detroit, MI, United States of America

* marinaf@eselx.ipl.pt

**Data Availability Statement:** The anonymized data are available through the OSF at https://osf.io/ymqcj.

## Abstract

Three infant regulatory behavior patterns have been identified during the Face-to-Face Still-Face paradigm (FFSF) in prior research samples: a *Social-Positive Oriented pattern (i.e.,* infants exhibit predominantly positive social engagement), a *Distressed-Inconsolable pattern (i.e.,* infants display conspicuous negative affect that persists or increases across FFSF episodes), and a *Self-Comfort Oriented* pattern (e.g., infants primarily engage in self-comforting behaviors such as thumb-sucking). However, few studies have examined these patterns outside US and European countries or evaluated potential cross-country differences in these patterns. In this study, we compared the regulatory behavior patterns of 74 Brazilian and 124 Portuguese infants in the FFSF at 3 months of age, and evaluated their links to demographic and birth variables. The prevalence of the three regulatory patterns varied by country. The most frequent pattern in the Portuguese sample was the Social-Positive Oriented, followed by the Distressed-Inconsolable and the Self-Comfort Oriented. However, in the Brazilian sample, the Distressed-Inconsolable pattern was the most prevalent, followed by the Social-Positive Oriented and the Self-Comfort Oriented. Moreover, in the Brazilian sample, familial SES was higher among infants with a Social-Positive pattern whereas 1st-minute Apgar scores were lower among Portuguese infants with a Distressed-Inconsolable Oriented pattern of regulatory behavior. In each sample, Social Positive pattern of regulatory behavior was associated with maternal sensitivity, Self-Comfort Oriented pattern of regulatory behavior with maternal control, and Distressed-Inconsolable pattern with maternal unresponsivity.

## Introduction

A growing body of research conducted in primarily US and European countries indicates that infants develop *organized patterns of regulatory behavior* during social interactions with their

**Funding:** This study was funded by the Portuguese Fundação para a Ciência e a Tecnologia/FEDER (PTDC/MHC-PED/1424/2014 - PI Marina Fuertes) and by the Brazilian FAPESP (2019/05593-6 e Coordenação de Aperfeiçoamento de Pessoal de Nível Superior - CAPES - PI Dionisia Lamônica).

**Competing interests:** No authors have competing interests.

caregivers [1–3]. These patterns are thought to reflect their repeated experiences of co-regulating distress and sharing affect with their caregivers, along with infants' own attempts to self-regulate and modulate arousal. Infant-caregiver dyads also co-create sensorimotor and affective "meanings" during social interactions that reflect the unique characteristics of their emerging relationship, and are hypothesized to contribute to the increasing complexity and coherence of the dyadic system [4].

Most studies have evaluated infants' early regulatory patterns during the Face-to-Face Still-Face paradigm (FFSF) [5], an experimental paradigm designed to evaluate infant and caregiver behavior before and after exposure to a social stressor (caregiver still-face). In studies using the FFSF with medically low-risk European or North American samples, the most predominant regulatory pattern displayed by infants is the *Social-Positive Oriented Pattern*, which is characterized predominantly by positive affect during high/moderate reciprocal interactions with their caregiver. Interactive mismatches are easily repaired, and infants tend to recover quickly from the stress of the still-face episode during the reunion episode [1, 3, 6]. The second most prevalent pattern in US and European samples is the *Distressed-Inconsolable Oriented* pattern. Infants displaying this pattern have periods of disengagement or moderate negative affect in the first episode and show exuberant negative affect in the still-face episode that continues or even increases during the reunion episode. The least prevalent regulatory pattern is the *Self-Comfort Oriented* pattern, in which infant's exhibit apparent avoidance of the adult (e.g., avoid eye contact) in the first and third interactive episodes, and engage in frequent self-comforting behaviors across all episodes, such as thumb-sucking [7–9].

Follow-up studies suggest that these early-emerging regulatory behavior patterns are relatively stable from 3 to 9 months of age [7] and are associated with variations in maternal interactive behavior in other contexts [8], and infants' attachment security at age one [10, 11]. Despite the apparent relevance of these patterns for infants' socioemotional development, very few studies have examined the distribution of these regulatory patterns outside US and European countries, or evaluated whether these patterns differ across countries or in different socio-demographic contexts. Among the few studies that have evaluated cross-country differences in infant behavior in the FFSF, most have focused on discrete infant behaviors (e.g., eye gaze, smiling) rather than organized regulatory patterns.

Findings from these studies suggest that infants' social-interactive behavior in the FFSF is shaped by their cultural and socio-demographic contexts. In several studies, both Chinese and Chinese American infants showed less negative affect [12, 13], more neutral affect [13], and less positive affect during the FFSF [12, 14, 15] than their European or European American/Canadian counterparts. Similar findings were reported in a longitudinal study of Japanese infants and their mothers [16]. In that study, Japanese infants did not display the classic still-face effect (i.e., decline of positive affect and increase of negative affect from baseline to the still-face episode followed by a partial recovery in the reunion episode) with respect to the interactive regulatory dimensions of gaze and positive affect see 11, for a review]. Overall, these studies suggest that infants with Chinese or Japanese heritage are less distressed and reactive across FFSF episodes than their European or European American/Canadian counterparts [17].

Other cross-country differences in infant discrete social-interactive behaviors during the FFSF have also been reported. In a comparison of 6-month-old Ghanaian and Canadian infants' responses in the FFSF, Ghanaian infants showed higher visual attention, more vocalizations, and smiling behaviors in the FFSF than age-matched Canadian infants [18]. The authors suggest that these differences may reflect variations in the prevalence of skin-to-skin contact in caregiving practices in the two countries. In an Ecuadorian study, mothers from a rural area who typically played with their four-month-old infants using contingent

responding, had infants with greater positive affect, whereas mothers who mostly engaged in attention-seeking play had infants who exhibited more negative affect [19].

In previous research, Brazilian infants born prematurely were more likely to exhibit conspicuous negative affect during the still-face episode of the FFSF at 5 months (corrected age), compared to their Portuguese, US or Swiss counterparts [3, 9, 20], and were more likely to experience greater difficulty recovering to baseline interactive values in the reunion episode, particularly when faced with higher maternal intrusiveness in the same recovery episode [21].

Overall, this cross-country literature suggests that discrete infant regulatory behaviors are embedded in contextual-environmental specifics such as variations in parenting, sociodemographics, and other social-cultural factors. However, few if any prior studies have evaluated cross-country differences in organized patterns of infant regulatory behavior during the FFSF, and their associations with these contextual-environmental factors.

## The present study

The first aim of the present study was to address this gap by comparing the distribution of the three previously identified patterns of infant regulatory behavior (Social-Positive Oriented, Distressed-Inconsolable, and Self-Comfort Oriented) at 3 months of age during the FFSF in a Brazilian and a Portuguese sample. It is important to understand whether these regulatory patterns of behavior are expressed in samples of infants from two countries with distinct sociodemographic and health characteristics.

Although Brazil and Portugal share the same language (Portuguese) and certain cultural traditions (e.g., religious), they vary on many other dimensions. Brazil is a large, diverse country with many socio-demographic and health disparities. For instance, the infant mortality rate in Brazil is high (about 13.8 deaths per 1,000 live births), and in the state of São Paulo (South-Eastern region), where the present study was conducted, the infant mortality rate is 10.7‰. Moreover, the average number of years of mothers' completed education in the São Paulo state is relatively low (8.85 years), compared with international standards [22]. In contrast, Portugal is a relatively small, and more culturally homogeneous country. Portugal is a member of the European Union, and has an excellent neonatal medical infrastructure, including free health, education, and social services for its citizens. Not surprisingly, Portugal has one of the world's lowest infant mortality rates (2.6%). Most of its citizens are from middle-class socioeconomic backgrounds. For example, Portugal requires 12 years of obligatory education, and the average rate of completed mandatory education for the general population is 75% [23]. Moreover, only 10% of the Portuguese population live in poverty, whereas 25% of the Brazilian population live in extreme poverty. Moreover, in past studies Portuguese mothers direct and respond more to infant solicitations with joy and verbally cues while Brazilian mothers are more likely to wait for infant reactions and respond to infant cues in a less intense manner (e.g., a low voice, whispering, nonverbal responses, and gentle touches) [16].

In light of these striking cross-country socio-economic, health and parenting differences, we expected to find significant differences in the distribution of the three infant regulatory patterns in each sample. Given the low level of social and health risks in the Portuguese culture, we hypothesized that the *Social-Positive Oriented* pattern of infant regulatory behavior would be more prevalent in the Portuguese sample than in the Brazilian sample.

A second aim was to examine whether the three infant regulatory patterns in each sample were associated with differences in (a) infants' proximal caregiving environment, as assessed via direct observations of maternal interactive behavior during mother-infant interactions; (b) infant and familial socio-demographic factors or (c) infant birth-clinical characteristics. Guided by findings in previous studies in our lab [8, 9], we expected that infants with the

Social-Positive Oriented pattern would be more likely to have mothers who were more behaviorally sensitive with them during mother-infant free-play interactions, and lower socio-demographic and birth-clinical risk indicators.

## Method

### Recruitment

Mother-infant dyads in Brazil and Portugal were recruited utilizing identical procedures during the same three-year time period. In both countries, trained research assistants contacted potential participants at metropolitan maternity hospitals located in Baúru, Sao Paulo (for Brazilian dyads) and in Lisbon or Oporto (for Portuguese dyads) and explained the study's purpose and procedures to them. Identical exclusion criteria were also used in recruiting each sample, including (a) evidence in the medical record or via maternal self-report of maternal substance (drug or alcohol) abuse during pregnancy; (b) evidence in the medical record or via maternal self-report of maternal mental health problems and/or chronic health conditions; (c) maternal age < 18 years at the time of the child's birth; (c) infants gestational age at birth < 36 gestational weeks; (d) presence of serious infant physical or health conditions (e.g., genetic disorders); or (e) the biological mother is not the infant's primary caregiver.

**Participants.** A total of 198 eligible mother-infant dyads (74 Brazilian and 124 Portuguese) gave their informed consent to participate in the study when the infants were 3 months old. Approximately half of the infants in each sample were female (57% in the Brazilian sample, 48% in the Portuguese sample). In both samples, all infants were healthy and clinically normal at delivery as determined by pediatric examination. All cases of sensory or neuromotor disabilities, serious illnesses, or congenital anomalies were excluded. None of the parents in either sample had any known drug/alcohol addiction or mental health problems. Dyads were primarily Brazilian-Caucasian or Portuguese-Caucasian in race/ethnicity, and participants within each sample varied in demographic and birth characteristics (Table 1).

Brazilian mothers were younger than the Portuguese mothers, had a lower level of completed education, were more likely to be from lower socioeconomic (SES) backgrounds, as ascertained with the National Social Economic Ranking Criteria (ABEP), were more likely to be employed outside the home, and had more children. Similarly, all infants in each sample were born at term or near-term, healthy and were equivalent in birth weight. However, Brazilian infants had a slightly lower mean gestational age at delivery and lower mean 1- and 5-minute Apgar scores than the Portuguese infants.

### Procedures

The aims and procedures of this study were approved by the Ethics Committees of all Health Units, Hospitals and Universities involved. Identical procedures were used in each sample and conducted according to the ethical guidelines stated in the Declaration of Helsinki. Mothers were informed about study aims, their participation rights, and written informed consent was obtained from all mothers before the study procedures began.

At 3 months postpartum, recruited mothers were recontacted to schedule a follow-up visit at a university laboratory. There, mother-infant dyads were first videotaped during a 5-minute free play session followed by the FFSF paradigm [5].

**Free play interactions.** The free play observations took place in calm and quiet laboratory settings and lasted 5 minutes. Mothers were instructed to play as they usually do at home, and were free to use toys or just engage in face-to-face interactions, including holding the infant or placing the infant in an age-appropriate chair.

**Table 1. Infant and family demographics.**

| | Brazil | | | Portugal | | | | |
|---|---|---|---|---|---|---|---|---|
| | *M* | *SD* | Min-Max | *M* | *SD* | Min-Max | *t* | *p* |
| Gestational weeks at birth | 38.73 | 1.31 | 36.06–41.0 | 39.54 | 1.08 | 37.00–41.57 | 5.265 | .001 |
| Birthweight (g) | 3253.91 | 447.50 | 2040–4380 | 3277.62 | 462.47 | 1790–4350 | .973 | ns |
| Apgar at first minute | 8.80 | 1.23 | 4–10 | 9.13 | .57 | 4–10 | 2.131 | .036 |
| Apgar at fifth minute | 9.76 | .46 | 8–10 | 9.98 | .16 | 9–10 | 3.691 | .001 |
| Number of siblings | 1.57 | .81 | 0–4 | .50 | .53 | 0–5 | 10.091 | .001 |
| Maternal age | 26.28 | 6.31 | 18–43 | 31.89 | 4.24 | 20–50 | 5–204 | .001 |
| Maternal years of education | 11.62 | 2.43 | 6–19 | 14.60 | 3.47 | 6–23 | 5.864 | .001 |
| | N | % | | N | % | | | |
| Infant sex | | | | | | | | |
| F | 42 | 56.8% | | 59 | 47.6% | | | |
| M | 32 | 43.2% | | 65 | 52.4% | | | |
| Maternal employment status | | | | | | | | |
| Employed | 39 | 52.7% | | 113 | 91.% | | | |
| Unemployed | 35 | 47.3% | | 11 | 8.9% | | | |
| Maternal SES | | | | | | | | |
| Upper | 4 | 5.5% | | 9 | 7.25% | | | |
| Middle | 33 | 39.1% | | 97 | 78.25% | | | |
| Lower | 37 | 55.4% | | 18 | 14.5% | | | |

Note. SES = socioeconomic status.

The videotapes of maternal behavior during free play were scored using the *Child-Adult Relationship Experimental Index* (CARE-Index) [24]. The CARE-Index assesses Maternal Sensitivity, Maternal Control, Maternal Unresponsivity, Infant Cooperative behavior, Infant Compulsivity, Infant Difficult behavior and Infant Passivity in seven aspects: facial expressions, verbal expressions, position and body contact, affection, turn-taking contingencies, control, and choice of activity. In this study, following the defined research aims, only the maternal scales were considered in the analysis.

Scoring of maternal interactive behavior was carried out separately in each sample using identical procedures. In each sample, the videotapes were scored by an independent team of two trained and reliable coders who were masked to the study's hypotheses and background characteristics. Intercoder agreement was determined separately in each sample using [all videos collected and included in the analysis (74 Brazilian and 124 Portuguese), and was calculated using. Results indicated (very good) agreement for all three maternal behavior dimensions in each sample (Brazil sample average ICC = .74; Portuguese sample average ICC = .81. After interrater reliability was calculated, discrepant ratings were resolved in conference.

**Face-to-Face Still-Face paradigm (FFSF) [5].** The FFSF consists of three successive two-minute episodes: (a) a baseline face-to-face interaction during which mothers were instructed to play with their infants as they would at home, albeit without using pacifiers or toys; (b) a still-face episode, during which mothers were asked to keep a neutral *"poker face"* while looking at the infants, and to refrain from talking, smiling, or touching the infant; and (c) a reunion episode, during which mothers were instructed to resume their normal play interaction with the infant [10].

Dyads were videotaped during the FFSF using two cameras: one focused on the infant's face and body and the other focused on the mother's face and upper torso. Both cameras were connected to an image mixer software system that generated a time-synchronized split-screen image of each partner on a single video record.

The *Coding System for Regulatory Patterns in the FFSF* [25] was used to score the videotapes of infants' behavior across the three FFSF episodes. This coding system describes three major behavior patterns of infant regulatory behavior: Social-Positive Oriented, Distressed-Inconsolable, and Self-Comfort Oriented. As described in [26], in order to assign each infant one of the three major categories (i.e., regulatory patterns), this coding system provides specific descriptions of infant behavior in each episode according to four behavioral dimensions: (a) behavior organization (e.g., the infant exhibits predominantly social positive behavior or distressful behavior or self-comforting behavior, or mixed behavior); (b) intensity of exhibited behavior (e.g., the infant displays prolonged and intense crying); (c) quality of behaviors (e.g., the infant reacts by displaying signals denoting pleasure such as smiles, laughter, and reciprocal neutral or positive vocalizations); and (d) infants' ability to recover from negative affect during the reunion episode of the FFSF. Each category is mutually exclusive. In classifying, the coder must decide which description best describes infant behavior across the three episodes, and assign that pattern to each case. Detailed information about each pattern description is presented in Table 2.

Scoring procedures used to classify the three infant regulatory patterns were identical in the Portuguese and Brazilian groups. In each sample, the videotapes were scored by an independent team of three trained and reliable coders who were masked to the study's hypotheses and background variables. Intercoder agreement was determined separately in each sample using all videos collected and included in the analysis (74 Brazilian and 124 Portuguese), and was calculated using Cohen's kappa coefficients. Results indicated very good agreement for all three regulatory patterns in each sample (Brazil sample average κ = .92; Portuguese sample average κ = .89). After interrater reliability was calculated, discrepant classifications were discussed and resolved in conference.

## Analytic plan

Potential group differences between the Brazilian and Portuguese samples on demographic and birth characteristics were tested using bivariate statistics (Table 1). The distribution of the three patterns of infant regulatory behavior in each sample was then obtained using univariate frequency analysis (Table 3). The association of the three infant regulatory patterns with maternal interactive behavior (Table 4) and demographic/birth variables was evaluated within each sample separately using one-way analyses of variance (ANOVA). The significance of differences between the three regulatory patterns was evaluated using Tukey post hoc tests. Cross-tabulation and chi-square analysis tested the associations between the three regulatory behavior patterns and infants' gender, maternal primiparous status, maternal employment status, and nationality. Data available in https://osf.io/ymqcj.

## Results

### Distribution of infant patterns of regulatory behavior in the Brazilian and Portuguese samples

The distribution of infant patterns of regulatory behavior in each sample is presented in Table 3.

The most common regulatory pattern observed for infants in the Brazilian sample was the Distressed-Inconsolable Oriented pattern (44.6%), followed by the Social-Positive Oriented (36.5%) and the Self-Comfort Oriented patterns (18.9%). In contrast, the distribution of regulatory patterns for infants in the Portuguese sample was consistent with those reported in other Portuguese samples [7]. The most common regulatory pattern for the Portuguese infants

**Table 2. Coding system for regulatory patterns in the FFSF.**

| Patterns of regulatory behavior | Description |
|---|---|
| *Social Positive Oriented* (Predominance of positive social behaviors and recover after still-face) | One of the three following options |
| | • Infants exhibit prolonged positive behaviors in the context of reciprocal interaction in the first episode. There is a clear and progressive decrease of positive affect during the still-face and a subsequent recovery during the third episode. Infants may take up to 30 seconds to recover in the last episode. |
| | • Infants exhibit a predominance of positive behaviors (but less frequent or less intense than last description) in the context of a reciprocal interaction. Nevertheless, a few periods of dyadic lack of synchrony can also be observed in the first episode. There is a progressive decrease of positive affect during the still-face and a subsequent recovery in the third episode. The recovery takes a maximum of 60 seconds. |
| | • Infants exhibit positive behaviors in a reciprocal interaction but there are often short or few long periods of lack of synchrony in the first episode, in which infants alternate with disturbance and self-comforting. Signs of disturbance and withdrawal may persist during the third episode, but infant gradually recover, and at least in last minute of this episode infants return to a reciprocal and positive interaction with their mothers. |
| *Distressed-Inconsolable* (Predominance of negative affect particularly in and after still-face, and failures in repairing interactive mismatches) | One of the two following options |
| | • Infants exhibit positive behavior during the first episode, but there are periods of disengagement or moderate negative affect. Infants react to the still-face with an increasing and persistent negative affect. Signs of disturbance and withdrawal persist in the third episode without recovering, although infants may present few or brief manifestations of interest. |
| | *Or*–Infants' engagement in the first episode alternates among periods of interest/attention, withdrawal, and active resistance/protest. Infants react to the still-face with prompt evident negative affect that persists or increases in the third episode. Infant distress is so intense that the researchers must end shortly the third episode. |
| *Self-Comfort Oriented* (Conspicuous avoidance in first and third episode and predominance of self-comfort during all episodes) | • Infants predominantly avoid contact, including gaze aversion, muscular tension when touched, and general discomfort without exhibition of evident negative affect (e.g., masked and rigid facial expression, restrained vocalizations) during the first and third episodes. Active resistance or protest are only occasional or briefly presented. During the second episode infants present predominantly self-comfort and exploring behaviors. Some infants seem more relaxed during the second episode compared to other episodes. Infants consistently use self-comforting behaviors across all episodes. |

Table 2 was adapted from [7] with author permission.

was the Social-Positive Oriented pattern (53.2%), followed by the Distressed-Inconsolable (35.5%) and the Self-Comfort Oriented patterns (13.3%).

## Association between maternal behavior and infant regulatory patterns

When qualitative dimensions of maternal interactive behavior during free play were compared across samples, several significant differences were observed. Maternal sensitivity was higher

**Table 3. Regulatory behavior patterns of infants during the FFSF at 3 months in the Brazilian and Portuguese samples.**

| Regulatory behavior patterns | Brazil | Portugal |
|---|---|---|
| Social-Positive Oriented (SPO) | 36.5% (n = 27; 1.0)[a] | 53.2% (n = 66; -1.3)[b] |
| Distressed-Inconsolable (DI) | 44.6% (n = 33; -.6)[a] | 35.5% (n = 44; .8)[a] |
| Self-Comfort Oriented (SCO) | 18.9% (n = 14;.-8)[a] | 13.3% (n = 14; 1.1)[a] |

Note. Pearson Chi-Square = 5.661; DF = 2, $p$ = .059; Identical superscript letter denotes categories whose column proportions do not differ significantly from each other

$p < .05$ (column proportions test with Bonferroni adjustment).

in the Portuguese sample than in the Brazilian sample [$t(2)$ = 2.697; $p$ = .01; IC (3.019–4.31); Portuguese sample $M$ = 9.55, $SD$ = 3.23; Brazilian sample $M$ = 7.83, $SD$ = 2.29]. Conversely, although not significant, maternal unresponsivity was slight higher in the Brazilian sample than in the Portuguese sample [$t(2)$ = 1.748, $p$ = .08; IC (.66–1.91); Brazilian sample $M$ = 2.75, $SD$ = 2.78; Portuguese sample $M$ = 1.63, $SD$ = 2.75]. No significant differences were found between the two samples for maternal control [$t(2)$ = .981, $p$ = .333; IC (.18–2.43); Brazilian sample $M$ = 3.43, $SD$ = 3.15; Portuguese sample $M$ = 2.80, $SD$ = 2.56].

However, similar within-sample associations between infant regulatory patterns in the FFSF and maternal interactive behavior during free play were observed in both samples, as tested with one-way ANOVAs (see Table 4). In each sample, infants with a Social Positive pattern of regulatory behavior were more likely to have mothers rated higher in sensitivity, compared to other infants. In contrast, infants classified with a Self-Comfort Oriented pattern of regulatory behavior were more likely to have mothers rated higher in control. In the Portuguese sample (but not the Brazilian sample), infants classified as Distressed-Inconsolable were more likely to have mothers rated higher in unresponsivity.

## Associations of demographic and birth characteristics with infant regulatory patterns

Results of one-way ANOVAs revealed no significant associations between the three regulatory behavior patterns and most sociodemographic and birth variables in either sample, including maternal age, education, number of siblings, infant birth weight, gestational weeks at delivery, and 5-minute Apgar score. Two exceptions were identified. In the Portuguese sample, infants' mean 1-minute Apgar score was lower among infants with a Distress-Inconsolable Oriented pattern of regulatory behavior, compared to infants with different regulatory patterns [$F(2)$ = 5.871;

**Table 4. Means, standard deviations, and ANOVA results for ratings of maternal behavior in free play at 3 months, according to patterns of infant regulatory behavior at 3 months.**

| Brazilian sample | Social-Positive Oriented M (SD) | Distressed-Inconsolable M (SD) | Self-Comfort Oriented M (SD) | F(2, 74) | p | Tukey HSD |
|---|---|---|---|---|---|---|
| Maternal variables Sensitivity | 9.93 (1.77)[a] | 6.29(1.53)[b] | 6.83 (1.85)[b] | 19.276 | .001 | a>b |
| Control | 1.71 (2.30)[a] | 3.94 (2.38)[a] | 5.50 (2.58)[b] | 5.848 | .006 | b>a |
| Unresponsivity | 2.36 (1.95) | 3.76 (3.36) | 1.67 (1.97) | 2.470 | .097 | |
| Portuguese sample | Social-Positive Oriented M (SD) | Distressed-Inconsolable M (SD) | Self-Comfort Oriented M (SD) | F(2, 99) | p | Tukey HSD |
| Maternal variables Sensitivity | 11.30(2.63)[a] | 6.32(3.19)[b] | 5.69 (2.72)[b] | 41.485 | .001 | a>b |
| Control | 1.64 (1.34)[a] | 3.03 (2.72)[a] | 6.77 (2.77)[b] | 33.216 | .001 | b>a |
| Unresponsivity | .46 (1.28)[a] | 3.50(2.78)[b] | .54 (2.77)[a] | 28.235 | .001 | b>a |

$p = .004$; Social-Positive $M = 9.28^a$, $SD = .49$; Distress-Inconsolable $M = 8.93^b$, $SD = .70$; Self-Comfort Oriented $M = 9.00^b$, $SD = .00$]. In contrast, in the Brazilian sample, familial SES was higher among infants with a Social-Positive pattern, compared with infants with either of the two other regulatory patterns [$F(2) = 16.817$; $p = .001$; Social-Positive $M = 5.20^a$, $SD = .87$; Distressed-inconsolable $M = 4.10^b$, $SD = 1.42$; Self-comfort oriented $M = 2.85^c$, $SD = 1.21$]. Cross-tabulation and chi-square analysis revealed no significant associations between the three regulatory behavior patterns and infants' gender or mothers' primiparous status, employment status, or nationality.

## Discussion

In this study, we observed cross-country differences in the distribution of infants' regulatory behavior patterns in the FFSF at 3 months of age. In the Portuguese sample, the Social-Positive Oriented pattern was the most prevalent pattern, followed by the Distressed-Inconsolable Oriented pattern and the Self-Comfort Oriented pattern. In contrast, for infants in the Brazilian sample, the most common pattern was the Distressed-Inconsolable Oriented pattern, followed by the Social-Positive Oriented pattern and the Self-Comfort Oriented pattern.

Given the striking current economic, health and social disparities between Portugal and Brazil, these cross-country behavioral differences are not surprising. For instance, the prevalence of the Social-Positive Oriented pattern was significantly higher in the Portuguese sample (53.2% in the Portuguese sample against 36.5% in the Brazilian sample). The distribution of early regulatory patterns observed for infants in the lower-risk Portuguese sample, characterized by higher SES and better health infrastructure, is consistent with that reported in several prior studies of infant behavior in the FFSF conducted in samples with similar sociodemographic characteristics in Portugal, the US, Canada, and some other European countries (e.g., Swiss) using this coding system or other classification methods/categories [3, 6, 7]. These studies show that the Social-Positive Oriented pattern (or the equivalent in other classification systems) is the most prevalent in these contexts. Infants with a Social-Positive Oriented pattern are more likely to engage in positive social interchanges in the context of reciprocal and positive interactions with their caregivers and are better able to soothe negative emotions following interactive mismatches (errors) triggered, for instance, by social stressors (e.g., a still-faced social partner). These infants are also more likely to re-engage in positive exchanges (i.e., reciprocal "serve and return" interactions with the caregiver during the reunion episode [27].

In contrast, in the higher-risk Brazilian sample, characterized by lower SES, younger maternal age, and higher caregiver burden, the Distressed-Inconsolable Oriented pattern was the most prevalent. This finding is consistent with those reported in other research evaluating Brazilian infants from S. Paulo in the FFSF [21]. Contrary to infants with a Social-Positive Oriented pattern, infants with a Distressed-Inconsolable pattern have difficulty in engaging in positive social interchanges with their caregivers, they react to moderate stressors with an increasing and persistent negative affect, not able to soothe their emotions.

The cross-country differences are not fully understood, they may reflect, in part, variations in infants' proximal caregiving environment. In the current study, we assessed caregiving quality using videotaped observations of maternal interactive behavior during mother-infant free play interactions. When cross-country differences in dimensions of maternal behavior were evaluated, Portuguese mothers were rated higher in sensitivity than Brazilian mothers, whereas Brazilian mothers were rated slightly higher in unresponsivity. Mothers in each sample did not differ significantly in control.

These findings are consistent with prior findings reported by our research team indicating that Brazilian mothers are more unresponsive and less sensitive to their infant's needs and

interests in the context of free play interactions at 3 months, compared to Portuguese mothers [9]. These findings also corroborate results from several other prior independent studies comparing maternal interactive behavior in Brazilian and Portuguese samples [21, 26]. In fact, cross-country differences in maternal interactive behavior are commonly reported in the literature. For example, Lowe et al. [28] found that European and US mothers, compared to Asian mothers, tend to vocalize more with their infants, engage in more eye contact, and respond more promptly to infant bids.

We speculate that these cross-country differences in infant regulatory patterns may reflect cross-country variations in mother-infant co-regulation. Several authors propose that infants gradually develop a capacity for emotion regulation via an infant-caregiver co-regulatory or a mutual regulatory system [4]. This system scaffolds infants´ immature regulatory skills and promotes the organization of dyadic regulatory patterns according to the successful dyadic reparation of interactive mismatches typically occurring in daily interactions [4, 29]. Probably infant and dyadic-specific regulatory patterns are the result of a complex equation in which maternal and infant interactive-social behavior are especially relevant.

In support of this idea, in the current sample, we found a similar pattern of *within-sample* associations between infant regulatory patterns in the FFSF and some dimensions of maternal interactive behavior during free play. In both the Brazilian and Portuguese samples, the Social Positive pattern of regulatory behavior was significantly associated with higher maternal sensitivity, whereas the Self-Comfort Oriented pattern was associated with higher maternal control. Moreover, the Distressed-Inconsolable pattern was associated with higher maternal unresponsivity, but this latter association reached statistical significance only in the Portuguese sample.

These findings mirror results from previous studies conducted by our research team showing that, in both Portuguese and Brazilian samples, maternal sensitivity is higher in infants with a Social-Positive Oriented pattern (or Positive Others oriented patterns in our original classification system) [7, 9, 26]. Therefore, the differences observed in these two samples may reflect country-specific variations in mother-infant interchanges that may contribute to dyadic successful interactions and infants' emerging regulatory patterns.

Another possible explanation for the cross-country differences in the distribution of infant regulatory patterns observed in the current study are country-specific differences in familial sociodemographic factors or infant health characteristics. In the Portuguese sample, for instance, the Social-Positive Oriented pattern was associated with a better infant health status at delivery, as reflected in higher Apgar scores at the first-minute post-birth. This finding deserves further evaluation because prior studies show that higher Apgar scores are associated with better maternal labor experiences and improved infant health outcomes [30], which, in turn, may reduce maternal anxiety and enhance mothers' ability to support their infants' emerging regulatory capacities and social communicative efforts.

In contrast, in the Brazilian sample, low SES was associated with the Distressed-Inconsolable pattern of regulatory behavior. A wealth of studies shows that inadequate financial and social resources, as marked by low SES, may heighten the risk for caregiving casualty, such as parental stress and anxiety, poor family support, distorted maternal representations of the infant, and harsh or neglectful parenting [31]. Many of these factors co-occur and are linked to poor infant regulation in the literature [32]. This may explain, in part, the high incidence of the Distressed-Inconsolable pattern in the current Brazilian sample.

Results from prior longitudinal studies show that the three infant regulatory patterns are stable from 3 to 9 months [7], affected by dyadic interactions [2, 8], maternal representations of infant behavior [9, 33], and infant birth status [2]. In the current cross-sectional study, findings regarding the associations of infant regulatory patterns with maternal interactive behavior, sociodemographics, and infant health were partially replicated in the Brazilian and

Portuguese samples. Although the direction of effects cannot be ascertained in the current study, our results suggest that, in both samples, organized patterns of infant regulatory behavior can be identified as early as 3 months of age, and that multiple proximal and distal contextual factors are associated with them.

## Limitations and strengths

A limitation of this study is the relatively small sample size, particularly in the Brazilian sample. This may have constrained the statistical power needed to identify other cross-group differences in infant regulatory patterns and their maternal, sociodemographic, and health correlates. For these reasons, findings from the present study should be viewed as preliminary and should be discussed in light of each sample's unique sociocultural and demographic characteristics.

Despite these limitations, results from this study contribute to the growing body of knowledge about infants´ early organized patterns of regulatory behavior during the FFSF, by shedding light on cross-country differences in the distribution of infant regulatory patterns that are already evident by 3 months of age. Findings also show that early regulatory patterns are associated with distinct demographic and birth variables across different countries.

## Author Contributions

**Conceptualization:** Marina Fuertes, Joana Gonçalves, Pedro Lopes dos Santos, Dionísia Aparecida Cusin Lamônica.

**Data curation:** Marina Fuertes, Camila da Costa Ribeiro, Miguel Barbosa, Ana Teresa Teodoro.

**Formal analysis:** Marina Fuertes.

**Funding acquisition:** Marina Fuertes, Dionísia Aparecida Cusin Lamônica.

**Investigation:** Marina Fuertes, Camila da Costa Ribeiro, Miguel Barbosa, Joana Gonçalves, Ana Teresa Teodoro, Dionísia Aparecida Cusin Lamônica.

**Methodology:** Marina Fuertes.

**Project administration:** Dionísia Aparecida Cusin Lamônica.

**Supervision:** Marina Fuertes, Marjorie Beeghly, Pedro Lopes dos Santos, Dionísia Aparecida Cusin Lamônica.

**Validation:** Marina Fuertes, Joana Gonçalves, Rita Almeida.

**Writing – original draft:** Marina Fuertes, Joana Gonçalves, Marjorie Beeghly.

**Writing – review & editing:** Marina Fuertes, Camila da Costa Ribeiro, Miguel Barbosa, Joana Gonçalves, Ana Teresa Teodoro, Rita Almeida, Marjorie Beeghly, Dionísia Aparecida Cusin Lamônica.

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
