## [Decision Letter · Decision Letter 0]

12 Mar 2021

PONE-D-20-40216

Patterns of Regulatory Behavior in the Still-Face Paradigm at 3 months: A Comparison of Brazilian and Portuguese Infants

PLOS ONE

Dear Dr. Fuertes,

Thank you for submitting your manuscript to PLOS ONE. After careful consideration, we feel that it has merit but does not fully meet PLOS ONE’s publication criteria as it currently stands. Therefore, we invite you to submit a revised version of the manuscript that addresses the points raised during the review process.

The reviewers provide detailed points that should be addressed in the revision. In particular, as suggested by the reviewers, it would be helpful to provide more information in the introduction about why differences are expected in the cultures under study. In addition, provide more details about the coding criteria for the classifying infant behavior patterns. I also agree with reviewer #2 that a review or assessment of mothers' behavior could be revealing. Please respond to each of the points and address each comment of the two reviewers in the revision.

We look forward to receiving your revised manuscript.

Kind regards,

Julie Jeannette Gros-Louis, PhD

Academic Editor

PLOS ONE

Journal Requirements:

3.We note that the grant information you provided in the ‘Funding Information’ and ‘Financial Disclosure’ sections do not match.

4. Please amend the manuscript submission data (via Edit Submission) to include author A., Silva.

5. Please amend your authorship list in your manuscript file to include author Rita Almeida.

6.We noticed you have some minor occurrence of overlapping text with the following previous publication(s), which needs to be addressed:

https://doi.apa.org/doiLanding?doi=10.1037%2Fdev0000616

https://www.tandfonline.com/doi/abs/10.1080/14616734.2020.1757730?journalCode=rahd20

In your revision ensure you cite all your sources (including your own works), and quote or rephrase any duplicated text outside the methods section. Further consideration is dependent on these concerns being addressed.

Reviewers' comments:

Reviewer's Responses to Questions

**Comments to the Author**

1. Is the manuscript technically sound, and do the data support the conclusions?

Reviewer #1: Partly

Reviewer #2: Yes

2. Has the statistical analysis been performed appropriately and rigorously? 

Reviewer #1: Yes

Reviewer #2: Yes

3. Have the authors made all data underlying the findings in their manuscript fully available?

Reviewer #1: Yes

Reviewer #2: Yes

4. Is the manuscript presented in an intelligible fashion and written in standard English?

Reviewer #1: Yes

Reviewer #2: Yes

5. Review Comments to the Author

Reviewer #1: The study compares Portuguese and Brazilian infants‘ response to the Still Face Task at 3 months. The Still Face Task has been used in many studies; but as the authors point out, most of these studies have been conducted in the US or Europe. Whether the paradigm elicits similar infant responses in non-western cultures is an important research question. Thus a cross-cultural study from a European and South American culture would be a welcome addition to the literature.

However, there are a number of gaps in the paper as it is currently written that limit its potential contribution to the literature. Most notably, (1) the authors need to make a case in the Introduction for why differences in infants’ reaction to the Still Face Task might emerge in the chosen cultures that would lead to hypotheses, (2) more information needs to be given concerning the coding system that results in the three infant patterns (Social-Positive, Distressed-Inconsolable, Self-Comfort).

Additional issues are outlined below.

Introduction

1. P. 2. Does the Distressed-Inconsolable pattern likely begin in the second episode (still face phase)? If so, make that clear.

2. P. 4. In the Chiodelli et al. (2020) study, was infants’ difficulty recovering from negative affect due particularly to maternal intrusiveness in the reunion episode or in the overall task?

3. Are you implying that the Portuguese sample would be an example of a US-European culture that has typically reported infants’ behavior in the Still Face Task and you are comparing it to a South American culture? Expand on the differences in the two cultures. Do you have hypotheses? If so, put them at the end of the Introduction with justification.

Method

4. P. 7. In the coding system, does behavior organization have one code per infant that is for the infant’s behavior over all three episodes of the Still Face Task ? If so, does an infant who is happy in the first episode but is unhappy in the second and third episodes get an overall unhappy code or a mixed code?

5. Pp. 7-8. Give more detail about the coding system. Were each of the dimensions coded on a scale or dichotomously (yes/no)? Is the quality of behaviors dimension just for positive emotional displays and intensity of exhibited behavior dimension just for negative emotional displays? How were the four dimensions combined into the six sub-categories? If the coding system has six sub-categories, are the three patterns (Social-Positive, Distressed-Inconsolable, Self-Comfort) part of the six sub-categories or are the six sub-categories collapsed into the three main styles? If the latter, how is this done?

Results

6. P. 9, third line from bottom. After “other national samples”, add references. Do you mean other European samples?

7. Table 2. The percentages are the most meaningful data in the table because the number of infants differed in the two samples. I suggest taking the percentages out of parentheses and presenting them first and putting the number of infants in parentheses. The note about the subscripts is confusing. I am not sure what is meant.

Discussion

8. P. 11, second paragraph. Consider putting this information into the Introduction as a means of forming hypotheses.

9. P. 12, second paragraph. Again, much of this information might be better placed in the Introduction leading to hypotheses.

10. P. 13. Although the authors allude to the cultural differences in infants’ responsive patterns may be due to differences in SES and infant health at birth, I suggest these possibilities be made more explicit. Portuguese mothers had higher SES than Brazilian mothers and their infants showed predominantly the Social-Positive pattern; Brazilian mothers with higher SES had infants who more likely showed the Social-Positive pattern. Likewise, Brazilian infants had lower Apgar scores than Portuguese infants and showed predominantly the Distress-Inconsolable pattern; Portuguese infants with lower Apgar scores more likely showed the Distress-Inconsolable pattern. Thus might SES and infant health at birth be possible explanation for the results?

11. P. 14, first paragraph. Is there evidence of these differences in parenting between Portuguese and Brazilian samples?

Minor

12. P. 4 middle of the page. Something is missing in the sentence beginning “The authors suggest, may reflect…”. Should it be changed to “The authors suggest that the differences reflect …”

13. P. 7, beginning of third paragraph. Change “one focuses” to “one focused”

Reviewer #2: Patterns of Regulatory Behavior in the Still-Face Paradigm at 3 months: A Comparison of Brazilian and Portuguese Infants

This study compared the regulatory behavior patterns in the Face-to-Face Stillface paradigm of 3 month old infants in two countries, Brazil and Portugal. As the authors point out there is need in the field for more research on the Stillface effects in various countries. It is important to know if the data gathered in the United States, where most of the research with this paradigm has occurred, can be replicated in other countries. So the goal of the current study is worthwhile. However, enthusiasm for the study is tempered by one major weakness. There may be several factors that need to be taken into account to explain a difference in how infants in other countries respond to the Stillface, including differences in number of caretakers, infant health or genetics. One important variable that may influence the infant’s response is the mother’s behaviour. This is especially true if the goal of a study is to understand differences between countries when the main difference is to be found in cultural differences. The authors state that they included video of the mother during the procedure but her behavior and interaction with the infant was not included in the data.

Other minor points:

1. There are not enough details about the procedural setup.

2. There should be information about how many coders and what percentage of videos were coded for reliability.

6. PLOS authors have the option to publish the peer review history of their article (what does this mean?). If published, this will include your full peer review and any attached files.

Reviewer #1: No

Reviewer #2: No

---

## [Author Response · Author response to Decision Letter 0]

27 Apr 2021

April 22, 2021

PONE-D-20-40216

Patterns of Regulatory Behavior in the Still-Face Paradigm at 3 months: A Comparison of Brazilian and Portuguese Infants

PLOS ONE

Dear Editor of PLOS ONE - Professor Julie Jeannette Gros-Louis

 The authors would like to thank the reviewers for their very helpful critiques and suggestions, and for allowing us the opportunity to revise our manuscript. We have carefully attended to, and have done our best to implement or address, each of your suggestions and comments. We have revised the manuscript in detail and edited according to each suggestion. 

Starting with the PLOS ONE editorial indications:

1. We ensured that the manuscript was formatted according PLOS ONE's style requirements. 

2. We included the title page within the main document listing all authors and affiliations.

3. We corrected the grant information provided in the ‘Funding Information’ and ‘Financial Disclosure’. Now, both sections match. 

4. We amended the manuscript submission data to include author Rita Almeida and included this author in the authorship list.

6. We reviewed all overlaps with our previous publications using the software URKUND, and included the respective citations or reworded the text.

7. We included our data set for open access in URL https://osf.io/ymqcj.

First reviewer

Overall Comments. We are grateful for the first reviewer for his/her significant suggestion and critiques. We agree with the suggestion regarding the introduction. Therefore, we reviewed the Introduction to explain why differences in infants’ reaction to the Still Face Task might emerge in the chosen countries and added study hypotheses. We hope the revised version of manuscript addresses better the cultural, economic, and social distinctions between both countries (in the introduction, present study and discussion) and their associations of each with infant regulatory patterns and maternal sensitivity. For that purpose, we also added new results for maternal sensitivity in both samples in Table 4 and we add new section of results “Association between maternal behavior and infant regulatory patterns”. We are also grateful for the suggestion about providing greater specification and information about the three infant patterns (Social-Positive, Distressed-Inconsolable, Self-Comfort) in the Methods section in table 2.

Also according to the reviewer suggestion:

Introduction

1. We clarified the information regarding the behavior of infants with Distressed-Inconsolable pattern across FFSF in page 4 last two sentences and page 5 first sentence. 

2. In the Chiodelli et al. (2020) study, infants’ had difficulty in recovering from negative affect due to maternal intrusiveness during the reunion episode. We detailed this result on page 6 - line 16.

3. Greater detail about specific differences and similarities between Brazil and Portugal are now provided in section Present Study - pages 6-8. 

Method

4. We provided more detail about the specific behaviors associated with the three infant regulatory patterns in each episode of the FFSF, and added a new table summarizing this information, on page 13 and 14. We agree that adding this information, the text is clearer and more exact.

5. We provided more detailed information about the coding system in pages , including the dimensions coded, quality of behaviors, and the six sub-categories, on page 13 (line 5-10; 10-16) and 14 (line 6-10) and in Table 2. 

Results

6. We clarified what by “other national samples” we meant Portuguese Samples. We corrected this error in page 16 line 15 and we included a reference. 

7. Table 3. We agree that the percentages of infants in each regulatory pattern are the most meaningful data in the table because the number of infants differed in the two samples. We accepted the suggestion to place the percentages first and put the number of infants in parentheses. We agree that this approach is clearer and easier to read in the table. 

8 and 9. We reframed the present study section to include the study hypotheses (page 8 line 5 and line 13).

10. The reviewer stated: “Although the authors allude to the cultural differences in infants’ responsive patterns may be due to differences in SES and infant health at birth, I suggest these possibilities be made more explicit. Portuguese mothers had higher SES than Brazilian mothers and their infants showed predominantly the Social-Positive pattern; Brazilian mothers with higher SES had infants who more likely showed the Social-Positive pattern. Likewise, Brazilian infants had lower Apgar scores than Portuguese infants and showed predominantly the Distress-Inconsolable pattern; Portuguese infants with lower Apgar scores more likely showed the Distress-Inconsolable pattern. Thus might SES and infant health at birth be possible explanation for the results?” We agree with this statement and have reframed substantially this section of discussion, to address the specific questions formulated.

11. We included additional evidence for differences in parenting between Portuguese and Brazilian samples on table 4, results section page 16 (starting in line 20) and page 18. Also these new results were discussed and introduced in the abstract. 

12. We changed “The authors suggest, may reflect…”. for “The authors suggest that the differences reflect …” in page 6 line 5.

13. We changed “one focuses” to “one focused, in page 12 line – 23 and 24.

 

Second reviewer

Overall suggestions. We are grateful for the reviewer for his/her appreciation of our manuscript and significant suggestion, particularly to add new results regarding maternal behavior in both samples. New results are now provided on table 4, results section page 16 (starting in line 20) and page 18. To explain the methodology used, we add a new section in methods named free play interactions. Also these new results were discussed and introduced in the abstract. . We believe that this new information helped us to better discuss the differences found between both samples and overall improved the manuscript. 

1. More detail information about the coding system (the dimensions coded, quality of behaviors, the six sub-categories) is presented in page 13 (line5-10; 16-19) and 14 (line 6-10) and in Table 2. 

2. We provided more detailed information about intercoder reliability for both the coding of maternal behavior during free play and the coding of infant regulatory patterns. We clarified that coding was carried out independently in each sample, by a team of trained reliable coders who were masked to the study’s hypotheses and background information. We also provided information about the number of coders in each team and what percentage of videos were coded for reliability by sample (see page 12 line 5-13 and page 14 line 10-14). We agree that adding this information makes the description of coding procedures more detailed and exact. 

Final comments. We also revised the manuscript for typos and errors. Since we included maternal behavior in the analyses we excluded three Brazilian mothers that had less than 18 years old and analyses were updated.

---

## [Editor Report · Decision Letter 1]

19 May 2021

Patterns of Regulatory Behavior in the Still-Face Paradigm at 3 months: A Comparison of Brazilian and Portuguese Infants

PONE-D-20-40216R1

Dear Dr. Fuertes,

We’re pleased to inform you that your manuscript has been judged scientifically suitable for publication and will be formally accepted for publication once it meets all outstanding technical requirements.

Kind regards,

Julie Jeannette Gros-Louis, PhD

Academic Editor

PLOS ONE
---

## [Editor Report · Acceptance letter]

25 May 2021

PONE-D-20-40216R1 

Patterns of Regulatory Behavior in the Still-Face Paradigm at 3 months: A Comparison of Brazilian and Portuguese Infants 

Dear Dr. Fuertes:

I'm pleased to inform you that your manuscript has been deemed suitable for publication in PLOS ONE. Congratulations! Your manuscript is now with our production department. 

Kind regards, 

on behalf of

Dr. Julie Jeannette Gros-Louis 

Academic Editor

PLOS ONE